

# Genetic variability and population structure of the Montezuma quail (*Cyrtonyx montezumae*) in the northern limit of its distribution

Eduardo Sánchez-Murrieta[1], Alberto Macías-Duarte[2],
Reyna A. Castillo-Gámez[3], Alejandro Varela-Romero[3],
Angel B. Montoya[4], James H. Weaver[5] and Nohelia G. Pacheco-Hoyos[3]

[1] Maestría en Biociencias, Universidad de Sonora, Hermosillo, Sonora, Mexico
[2] Cuerpo Académico de Recursos Naturales, Universidad Estatal de Sonora, Hermosillo, Sonora, Mexico
[3] Departamento de Investigaciones Científicas y Tecnológicas, Universidad de Sonora, Hermosillo, Sonora, Mexico
[4] Partners for Fish and Wildlife Program, United States Fish and Wildlife Service, Las Cruces, New Mexico, United States of America
[5] Texas Parks and Wildlife Department, Fort Davis, Texas, United States of America

Corresponding author
Alberto Macías-Duarte,
alberto.macias@ues.mx

## ABSTRACT

Restricted movement among populations decreases genetic variation, which may be the case for the Montezuma quail (*Cyrtonyx montezumae*), a small game bird that rarely flies long distances. In the northern limit of its distribution, it inhabits oak-juniper-pine savannas of Arizona, New Mexico, and Texas. Understanding genetic structure can provide information about the demographic history of populations that is also important for conservation and management. The objective of this study was to determine patterns of genetic variation in Montezuma quail populations using nine DNA microsatellite loci. We genotyped 119 individuals from four study populations: Arizona, Western New Mexico, Central New Mexico, and West Texas. Compared to other quail, heterozygosity was low ($\bar{H}_0 = 0.22 \pm 0.04$) and there were fewer alleles per locus ($\bar{A} = 2.41 \pm 0.27$). The global population genetic differentiation index $R_{ST} = 0.045$ suggests little genetic structure, even though a Bayesian allocation analysis suggested three genetic clusters ($K = 3$). This analysis also suggested admixture between clusters. Nevertheless, an isolation-by-distance analysis indicates a strong correlation ($r = 0.937$) and moderate evidence ($P = 0.032$) of non-independence between geographical and genetic distances. Climate change projections indicate an increase in aridity for this region, especially in temperate ecosystems where the species occurs. In this scenario, corridors between the populations may disappear, thus causing their complete isolation.

## INTRODUCTION

Geographical patterns of genetic variation are influenced by a species ability to disperse (*Waters et al., 2020*) and low dispersal ability may reduce local genetic variation in isolated populations (*Frankham, 1996*). If dispersal is limited, then the concomitant reduced genetic diversity can limit local adaptability (*Allendorf & Luikart, 2007*), and may result in a detriment of the long-term survival of the metapopulation (*Arif & Khan, 2009*). Similarly, genetic variation loss reduces population viability by decreasing average individual fitness (*Reed & Frankham, 2003*). Population size reductions in harvested wildlife species have disrupted their geographical distribution, increased their isolation, and reduced gene flow (*Allendorf et al., 2008*). Under this metapopulation dynamic scenario, persistent harvest may further exacerbate the loss of genetic variability (*Harris, Wall & Allendorf, 2002*). The Montezuma quail (*Cyrtonyx montezumae* Vigors 1830) is a small game bird with limited dispersal by flight, that has a naturally disjointed geographic distribution. Montezuma quail inhabit temperate woodlands and savannas associated with mountains ranges in southeastern Arizona, southern New Mexico, and west Texas in the United States (*Stromberg, Montoya & Holdermann, 2020*). The Montezuma quail's geographic distribution extends southwards into Mexico along the Sierra Madre Occidental and Oriental to the Trans-Mexican Volcanic Belt and the Sierra Madre del Sur. Montezuma quail often occupy habitat patches in isolated mountain ranges that are widely separated by unsuitable arid lands (*Stromberg, Montoya & Holdermann, 2020*). Dense grass is a necessary habitat component for cover (*Brown, 1979*), nesting (*Bishop, 1964*), and forage (*López-Bujanda et al., 2022*). Montezuma quail are sedentary, with home range sizes averaging 50 ha (*Chavarria et al., 2017*). The Montezuma quail nesting season is from June to October and is largely influenced by summer precipitation. Females produce a single clutch of around 10 eggs (*Stromberg, Montoya & Holdermann, 2020*). Montezuma quail live in coveys during the non-breeding season, but form pairs for nesting by as early as late February. The species feeds primarily on underground plant structures, such as bulbs and tubers of woodsorrel (*Oxalis* spp.), and sedge (*Cyperu*s spp.), but also feed on acorns (*Quercus* spp.), and a large variety of seeds and insects (*López-Bujanda et al., 2022*).

The Montezuma quail has a patchy distribution with numerous disjunct populations in Arizona, New Mexico, and Texas (*Stromberg, Montoya & Holdermann, 2020*). This habitat fragmentation may have an additive effect with genetic drift on the loss of genetic variation of isolated populations. For instance, naturally fragmented habitat has led to reduced levels of genetic variation in the Mexican spotted owl (*Strix occidentalis lucida*) in southeastern Arizona. Differences in management by state agencies may have also affected quail genetic structure. Montezuma quail are hunted in Arizona and New Mexico (*Stromberg, Montoya & Holdermann, 2020*; *Heffelfinger & Olding, 2000*), but not in Texas, where populations are restricted to the Trans-Pecos region and the Edwards Plateau (*Albers & Gehlbach, 1990*). Furthermore, geographic genetic variation of Texas populations might have been disrupted in the mid 1970s when a series of reintroductions occurred in west and central Texas using quail from Arizona (*Wauer, 1973*; *Stromberg, Montoya & Holdermann, 2020*).

The success of these reintroductions, and their contribution to the gene pool in those populations, has not been confirmed (*Armstrong, 2006*).

A population genetic survey is a convenient way to evaluate the effect of isolation, fragmentation, management, and reintroductions on the viability of Montezuma quail populations. The objective of this study was to determine the patterns of genetic variation among Montezuma quail populations in Arizona, New Mexico and Texas using DNA microsatellite markers.

## MATERIALS AND METHODS

Tissue samples were extracted from specimens hunted in Arizona and New Mexico under numerous hunting licenses issued by Arizona Game and Fish Department and New Mexico Department of Game and Fish. Tissue samples from Texas originated from specimens collected under Scientific Permit Number SPR-0410-139 issued by Texas Parks and Wildlife Department. We initially allocated 119 individual samples to four populations: Arizona (AZ), Central New Mexico (CNM, east of the Río Grande), Western New Mexico (WNM, west of the Río Grande) and West Texas (WTX). These geographic designations produced an unbalanced sample size, with a relatively low sample size allocated to CNM ($n = 12$ individuals). Since unbalanced sampling affects the inference of population structure (*Meirsman, 2019*), we re-allocated 14 samples from the two easternmost collection locations of the WNM population to the CNM population. Ultimately, we allocated the 119 samples as follows: 32 to AZ, 36 to WNM, 26 to CNM and 25 to WTX (Fig. 1). We used 25 mg of muscle tissue from the right wing to extract genomic DNA using a Qiagen® DNeasy Blood and Tissue extraction kit.

Twenty DNA microsatellite loci developed by *Schable et al. (2004)* for *Colinus virginianus* were evaluated. The choice of these was made based on heterozygosity and the number of alleles reported by the authors. For microsatellite amplification, 25 μL were prepared for PCR reactions whose final concentrations were 2 μL of genomic DNA (with a concentration of 50/μL), 12.5 μL of MasterMix (GoTaq® Colorless Master Mix, Promega), 1 μL of each oligo and 8.5 μL of PCR water without endonucleases. Thermocycler conditions for the amplification were modified according to the MasterMix manufacturer specifications and the alignment temperatures of the oligos, which varied between 45 °C and 57 °C. The program used was as follows: a cycle of 95 °C for 2 min; five cycles of 94 °C for 30 s, 45 °C for 30 s and 72 °C for 30 s; then 35 cycles of 94 °C for 45 s, 45 °C for 45 s and 72 °C for 1 min; followed by a cycle of 72 °C for 2 min. Ten microsatellite loci with the highest polymorphism and concordance in the fragment size as reported by the author were chosen. Selected loci were marked with fluorophores in the sequence 5′–3′. A post-PCR multiplex array for fragment reading was performed. This was performed on an ABI 3730xl sequencer (Applied Biosystems, Waltham, MA, USA) of Macrogen. We used GeneScan™ 350 ROX™ dye Size Standard for sizing DNA fragments. Allele scoring was conducted using program GeneMarker v. 2.6.4 (*Hulce et al., 2011*). We did not run fragment analyses on individual samples more than once and we could not estimate error rates associated with allele scoring. Loss of alleles, null alleles and excess homozygotes were assessed using program Micro-Checker v 2.2.3 (*Van Oosterhout et al., 2004*). Genetic
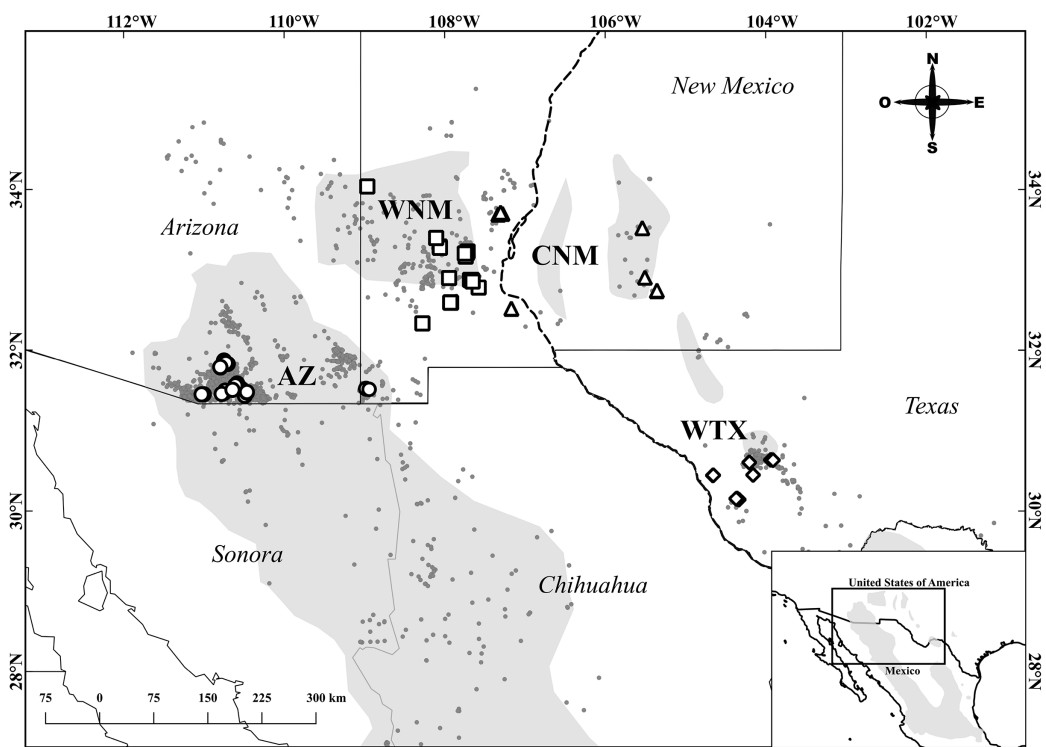

**Figure 1 Sampling locations of Montezuma quail.** The gray area represents the estimated geographic distribution of the Montezuma quail (*BirdLife International, 2016*). Circles, squares, triangles, and diamonds represent individuals from the populations of Arizona (AZ), Western New Mexico (WNM), Central New Mexico (CNM), and West Texas (WTX), respectively. Gray dots are records from e-bird (*Sullivan et al., 2009*) and GBIF (*GBIF.org, 2023*). State division for the United States by *National Weather Service (2023)*. State division for Mexico by *Instituto Nacional de Estadística y Geografía (2022)*.                

variability estimator's locus alleles ($A$) expected heterozygosity ($H_e$) and observed heterozygosity ($H_o$) were obtained for each locus and population using GenAlEx v 6.5 (*Smouse & Peakall, 2012*). Using the Arlequin v 3.5 software (*Excoffier & Lischer, 2010*), possible deviations from the Hardy-Weinberg equilibrium (HWE), were calculated for each locus in each population. Linkage disequilibrium for each pair of loci across populations was tested using the Fisher's method as implemented by Genepop on the Web (*Raymond & Rousset, 1995*; *Rousset, 2008*).

Population genetic differentiation index $R_{ST}$ was calculated using Arlequin v 3.5 (*Excoffier & Lischer, 2010*) to determine the degree of genetic differentiation among populations. Population structure analysis was conducted using software STRUCTURE (*Pritchard, Stephens & Donnelly, 2000*), which estimates the posterior probability of the data given existence of $K$ clusters or groups under Hardy–Weinberg equilibrium and estimates the individuals' posterior membership probability to each of the $K$ clusters. Individuals are then assigned to the cluster that holds the highest probability. Parameters established for the analysis were 10,000 burnins, 50,000 repetitions of Monte Carlo Markov Chains, 25 iterations, a $K$ value (number of clusters) between 1–4, an ancestral mixing model, and a correlated allelic frequency model. Outputs from program STRUCTURE

were analyzed using the method of *Evanno, Regnaut & Goudet (2005)* as implemented by software STRUCTURE HARVESTER (*Earl, 2012*) to determine the most probable number of genetic clusters. This method estimates Delta $K$ ($\Delta K$), which is the difference between the values of the logarithmic likelihood in each analysis iteration for the four clusters.

A Mantel test was conducted using program Genepop 4.3 (*Rousset, 2008*) to determine if there was a distance isolation pattern among Montezuma quail populations. Statistic $R_{ST}$ was used as a measure of genetic distance between populations. We performed 10,000 permutations to estimate the statistical significance ($P < 0.05$) of the null hypothesis of independence between genetic and geographic distance.

# RESULTS

A reduced number of DNA microsatellite loci out of the 20 loci tested were used for genetic analyses. We use locus (Quail) names as in *Schable et al. (2004)* hereafter. Loci Quail 3, Quail 13, Quail 24, and Quail 27 did not amplify for some individuals. For the first three loci, only one individual did not amplify in CNM, WNM, and AZ respectively. For Quail 27, no alleles were recovered in three individuals of CNM and two of WNM. For Quail 41, despite amplification of fragments in agarose gels, we could only genotype 21 individuals and this locus was excluded from analyses. Null alleles were detected in Quail 27 and Quail 44, and these were excluded from the analyses of genetic structure (see below). Seven loci were polymorphic in at least one population, while Quail 25 and Quail 44 were monomorphic for all populations (Table 1). We found private alleles for Quail 03, Quail 13, Quail 27, and Quail 31 in populations AZ, CNM and WTX. Quail 27 deviated from HWE in AZ, CNM and WNM, and had heterozygous deficit. Quail 31 also deviated from HWE in all populations, with an excess of heterozygotes in AZ and deficit in the other three populations. Quail 24 did not meet HWE at WNM, where many individuals had the same homozygous genotype. There was suggestive evidence of linkage disequilibrium for only pairs Quail 9–Quail 31 ($\chi^2 = 9.58$, d.f. = 4, $P = 0.048$) and Quail 27–Quail 31 ($\chi^2 = 16.86$, d.f. = 8, $P = 0.032$) and no evidence of linkage disequilibrium for the other 23 pairs ($P > 0.05$). Hence, no locus was discarded due to linkage disequilibrium. In the end, all samples ($n = 119$) were analyzed using 9 DNA microsatellite loci.

Overall estimators of genetic variability had the following values: mean number of alleles $2.41 \pm 0.27$ per locus (range = 1–11), mean observed heterozygosity of $\bar{H}_O = 0.22 \pm 0.04$ and mean expected heterozygosity of $\bar{H}_E = 0.24 \pm 0.04$ (Table 1). Genetic variability estimates remained similar among populations. CNM had the greatest observed heterozygosity ($\bar{H}_O = 0.25 \pm 0.08$), followed by WTX ($\bar{H}_O = 0.23 \pm 0.08$), and AZ ($\bar{H}_O = 0.22 \pm 0.10$), while the WNM population had the lowest ($\bar{H}_O = 0.19 \pm 0.07$).

Overall, among-population genetic differentiation was low ($R_{ST} = 0.045$). However, pairwise differentiation between distant populations AZ–WTX ($R_{ST} = 0.094$. $P = 0.001$), AZ–CNM ($R_{ST} = 0.061$. $P = 0.009$), and WTX–WNM had statistical significance ($R_{ST} = 0.094$, $P = 0.001$) (Table 2). We found three genetic clusters ($K = 3$) (Fig. 2), whose membership probabilities had a west-east gradient (Fig. 3); study population geographic positions was not an input of the STRUCTURE analysis. Most individuals from the AZ (66%) had a higher probability of membership in one cluster (Arizona cluster hereafter,

**Table 1 Genetic variability estimators for nine microsatellite loci of Montezuma quail populations in Arizona, Central New Mexico, Western New Mexico, and West Texas.**

| | Arizona | | | | Central New Mexico | | | | Western New Mexico | | | | West Texas | | | |
|---|---|---|---|---|---|---|---|---|---|---|---|---|---|---|---|---|
| Locus | N | A | $H_o$ | $H_e$ | N | A | $H_o$ | $H_e$ | N | A | $H_o$ | $H_e$ | N | A | $H_o$ | $H_e$ |
| Quail 03 | 32 | 2 | 0.16 | 0.2 | 23 | 3 (1) | 0.22 | 0.20 | 36 | 3 | 0.22 | 0.20 | 25 | 2 | 0.12 | 0.11 |
| Quail 09 | 32 | 1 | 0.00 | 0.00 | 24 | 2 | 0.13 | 0.12 | 36 | 2 | 0.17 | 0.15 | 25 | 1 | 0.00 | 0.00 |
| Quail 13 | 32 | 1 | 0.00 | 0.00 | 24 | 1 | 0.00 | 0.00 | 35 | 1 | 0.00 | 0.00 | 25 | 2 (1) | 0.04 | 0.04 |
| Quail 14 | 32 | 2 | 0.41 | 0.33 | 24 | 2 | 0.58 | 0.50 | 36 | 2 | 0.47 | 0.49 | 25 | 2 | 0.56 | 0.50 |
| Quail 24 | 31 | 3 | 0.42 | 0.35 | 24 | 3 | 0.42 | 0.40 | **36** | **2** | **0.03** | **0.08** | 25 | 3 | 0.36 | 0.30 |
| Quail 25 | 32 | 1 | 0.00 | 0.00 | 24 | 1 | 0.00 | 0.00 | 36 | 1 | 0.00 | 0.00 | 25 | 1 | 0.00 | 0.00 |
| Quail 27 | **32** | **3** | **0.19** | **0.45** | **22** | **4 (2)** | **0.27** | **0.61** | **33** | **2** | **0.21** | **0.5** | 25 | 4 (1) | 0.52 | 0.62 |
| Quail 31 | **32** | **5 (1)** | **0.84** | **0.63** | **24** | **7 (1)** | **0.63** | **0.77** | **36** | **6** | **0.58** | **0.68** | **25** | **7 (3)** | **0.48** | **0.69** |
| Quail 44 | 32 | 1 | 0.00 | 0.00 | 24 | 1 | 0.00 | 0.00 | 36 | 1 | 0.00 | 0.00 | 25 | 1 | 0.00 | 0.00 |
| Mean | 31.89 | 2.11 | 0.22 | 0.22 | 23.89 | 2.67 | 0.25 | 0.29 | 35.56 | 2.22 | 0.19 | 0.23 | 25.00 | 2.56 | 0.23 | 0.25 |
| S.E. | 0.11 | 0.45 | 0.10 | 0.08 | 0.35 | 0.65 | 0.08 | 0.10 | 0.34 | 0.52 | 0.07 | 0.09 | 0.00 | 0.65 | 0.08 | 0.09 |

**Note:**
Symbols $N$, $A$, $H_o$ and $H_e$ denote sample size, number of alleles (exclusive alleles), observed heterozygosity and expected heterozygosity, respectively. Numbers marked in bold correspond to loci that deviated from Hardy-Weinberg equilibrium under Fisher's exact test ($P < 0.05$).

**Table 2 Genetic differentiation index values $R_{ST}$ (below the diagonal) between Montezuma quail populations in Arizona (AZ), Central New Mexico (CNM), Western New Mexico (WNM), and West Texas (WTX).**

| Population | AZ | CNM | WNM | WTX |
|---|---|---|---|---|
| AZ | – | **0.009** | 0.054 | **0.001** |
| CNM | 0.061 | – | 0.063 | 0.072 |
| WNM | 0.014 | 0.021 | – | **0.009** |
| WTX | **0.094** | 0.041 | **0.059** | – |

**Notes:**
Numbers above the diagonal represent $P$-values for each pairwise population comparison from an exact Fisher test. The index $R_{ST}$ values that were statistically significant ($P < 0.05$) are marked in bold.

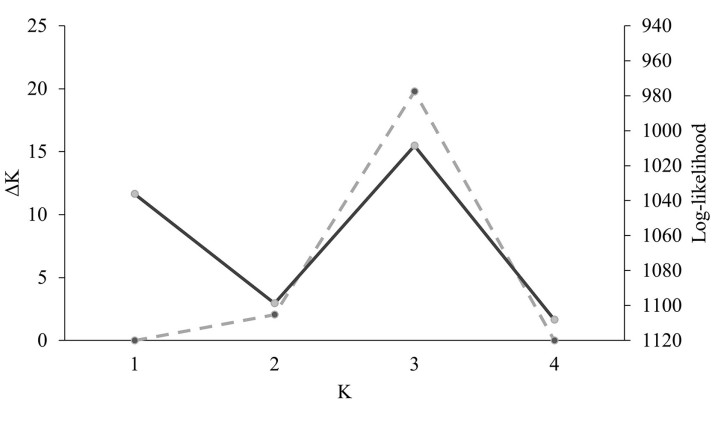

**Figure 2 Log-likelihood values *vs.* number of groups from the Bayesian method obtained in program STRUCTURE (*Pritchard, Stephens & Donnelly, 2000*) ($\Delta K$) for DNA microsatellite data of Montezuma quail from Arizona, New Mexico, and Texas.**

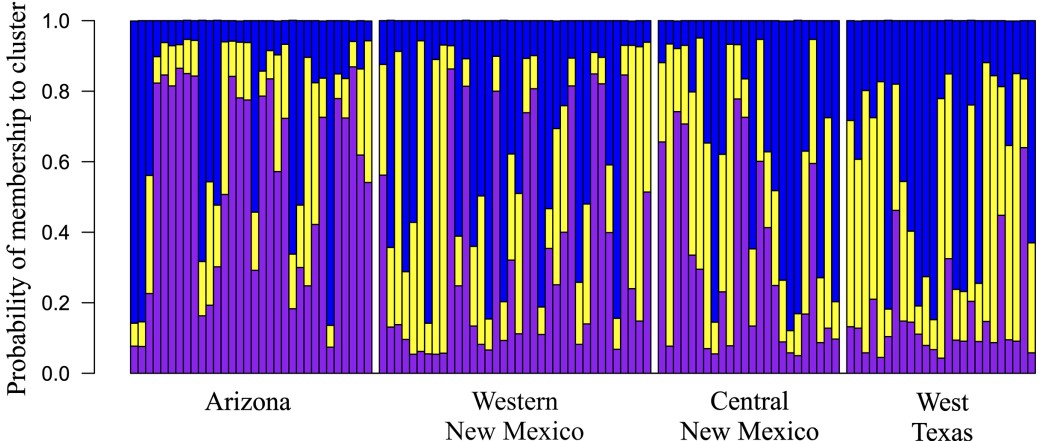

**Figure 3 Geographic variation among Montezuma quail populations in posterior membership probability in each of the clusters inferred by program STRUCTURE (*Pritchard, Stephens & Donnelly, 2000*), showing that geographic regions (*i.e.* study populations) moderately predict genetic clusters.** Each stacked bar represents an individual's ancestry estimates of each genetic cluster (Arizona, Texas, and New Mexico). Probability of individual membership is indicated by color (purple–Arizona, blue–New Mexico, yellow–Texas).

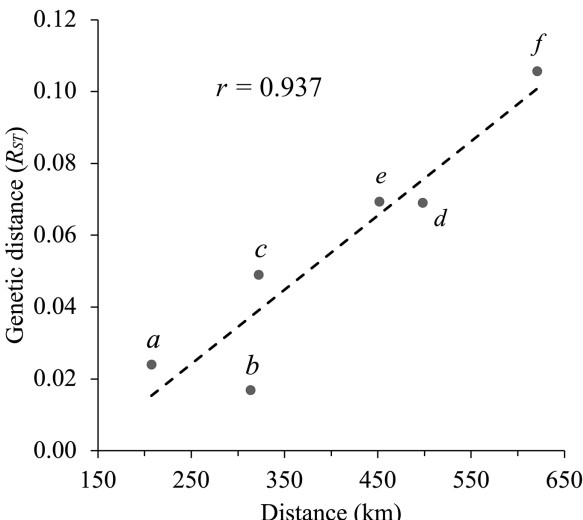

**Figure 4 Correlation between geographical distance and the $R_{ST}$ statistic values for each pair of populations (Mantel test, $r = 0.937$, $P = 0.032$ based on 10,000 permutations).** The letters on the graph indicate each pairwise population comparison: (*a*) CNM–WNM, (*b*) AZ–WNM, (*c*) CNM–WTX, (*d*) AZ–CNM, (*e*) WNM–WTX, and (f) AZ–WTX.

Fig. 3). The within-individual probability of membership in this Arizona cluster declined towards the east, with only 12% of the WTX individuals assigned to the Arizona cluster. Likewise, most individuals from WTX (48%) had a higher probability of membership in a second cluster (Texas cluster hereafter, Fig. 3). The probability of membership in the Texas cluster within individuals declined towards the west, with only 3% of the AZ individuals assigned in the Texas cluster. However, the probability of membership in the third cluster

(New Mexico cluster hereafter, Fig. 3) had no geographic gradient (Fig. 3). WNM (47%) and WTX (40%) individuals were often assigned to the New Mexico cluster.

Geographical distance and genetic distance ($R_{ST}$) were strongly correlated across populations ($r = 0.937$) (Fig. 4). Nevertheless, we found only moderate evidence (Mantel test, $P = 0.032$ based on 10,000 permutations) of an association between geographical distance and genetic distance. According to $R_{ST}$ statistics, AZ and WTX populations had the greatest genetic and geographic distances, while CNM and WNM populations had a shorter genetic and geographical distance.

## DISCUSSION

Montezuma quail populations appear to be effectively isolated from each other over relatively short geographical distances in the southwestern USA. Nevertheless, our results and their discussion should be taken with caution due to the relatively low number of loci analyzed (nine loci). Low heterozygosity ($\bar{H}_O = 0.22 \pm 0.02$), few alleles per locus ($A = 2.41 \pm 0.27$) and a high proportion of fixed loci (3 out of 9 loci) demonstrate low genetic variability of the Montezuma quail in the southwestern United States. For instance, *DeYoung et al. (2012)* reported an average observed heterozygosity of $H_o = 0.58$ in seven DNA microsatellite loci for bobwhite (*Colinus virginianus*) populations in southern Texas. *Orange et al. (2014)* reported a $H_o$ from 0.250–0.928 and an average $A = 7$ in 23 DNA microsatellite loci for scaled quail (*Callipepla squamata*) populations in Arizona, Colorado, Oklahoma, and Texas. Similarly, *Mathur et al. (2019)* reported low levels of genetic variability in the Montezuma quail populations in Arizona, New Mexico and West Texas using genome-wide single nucleotide polymorphism, with $H_o = 0.32 \pm 0.17$. Low genetic variability of Montezuma quail populations may be caused by drift in small and isolated populations inhabiting oak-pine-juniper savanna islands separated by vast desert grasslands. In this regard, low migration rates between the peripheral populations and those closer to the center of the geographic range could promote gene drift (*Lesica & Allendorf, 1995*), where fixed alleles and reduction of genetic variability are more likely to occur. For instance, the Mexican spotted owl had reduced levels of genetic (mtDNA) variation in the Madrean sky islands of southeastern Arizona (co-inhabited by Montezuma quail) compared to the remaining owl populations in Arizona, New Mexico, Utah, and Colorado (*Barrowclough et al., 2006*).

Previous genetic work (*Allen, 2003*) found that Montezuma quail from Arizona and Texas were not genetically distinct from one another based on mitochondrial DNA sequences. In this study, slight population structure was shown through the low $R_{ST}$. Nonetheless, pairwise $R_{ST}$ values indicate a low to moderate differentiation in some study populations. For instance, AZ is different from CNM ($R_{ST} = 0.061$; $P = 0.011$) and WTX ($R_{ST} = 0.094$; $P = 0.001$). In addition, AZ is not differentiated from WNM, suggesting an extant or recent corridor for Montezuma quail between these two study populations. Montezuma quail occur through the forested Mogollon Rim, which may connect WNM to central Arizona and to mountain ranges to southern Arizona (AZ) as both eBird sightings (*Sullivan et al., 2009*) and GBIF (*GBIF.org, 2023*) records suggest (Fig. 1). The Montezuma quail's ability to disperse this distance may not be as limited as presumed. A radiotelemetry

study conducted in Texas found that Montezuma quail move up to 15 km (*Greene et al., 2020*). Furthermore, the lowest genetic differentiation occurred between neighboring WNM and CNM ($R_{ST}$ = 0.014; $P$ = 0.063). Although some CNM individuals appear to be geographically isolated from the rest of the population (for example, individuals collected in the Lincoln National Forest), Montezuma quail can occur in areas with riparian vegetation near oak forest (*Stromberg, Montoya & Holdermann, 2020*), hence, they may be using riparian corridors to transit between the WNM and CNM populations. Sightings on the Rio Grande (*Sullivan et al., 2009*) also suggest connectivity between WNM and CNM through this river.

Despite a low genetic differentiation among our study populations, patterns of genetic variation from all our analyses suggests isolation by distance, which arises from limited geneflow between distant populations (*Wright, 1943*). $R_{ST}$ values between distant populations AZ and WTX ($R_{ST}$ = 0.094; $P$ = 0.001) and WTX–WNM populations ($R_{ST}$ = 0.059; $P$ = 0.013) were the highest among all pairwise comparisons. In contrast, the previous mtDNA survey by *Allen (2003)* found no genetic differentiation between Montezuma quail populations in Arizona and Texas. Differentiation between neighboring WTX–CNM populations ($R_{ST}$ = 0.033) was suggestive ($P$ = 0.096) that gene flow occurs between these populations >200 km apart. This area is approximately 20 km away from the Guadalupe Mountains, which is near the Texas–New Mexico state line, thus representing a possible route of exchange between the two study populations. However, the current existence of a large extension of unsuitable arid land between the two mountainous areas and the $R_{ST}$ may suggest recent divergence. Furthermore, our Bayesian analysis suggested three clusters, whose assignment probabilities among Montezuma quail followed a longitudinal gradient (Fig. 3), although mixing between clusters was evident. However, a Bayesian assignment by *Mathur et al. (2019)* did not detect separation between Arizona and New Mexico using genomic-wide single nucleotide polymorphism. Nevertheless, *Mathur et al. (2019)* also found lower differentiation between the populations of AZ–WNM and WNM–CNM. Thus, these populations may represent a single cluster, while the WTX population would be isolated from the rest. Approximately 40% of WTX Montezuma quail were assigned to the NM Cluster, suggesting recent admixture between clusters. Results from this work support the idea, proposed by *Mathur et al. (2023)*, that the Montezuma quail is a "ring species", where AZ and WTX populations were colonized by divergent populations dispersing northbound from the ancestral population in central Mexico through the Sierra Madre Occidental and Sierra Madre Oriental, respectively. Alternatively, our results may also suggest gene flow has recently stopped. Connectivity between Montezuma quail populations in the American southwest and northern Mexico may have been more widespread before the late 19[th] century when extensive ranching began reducing and depleting the grass cover (*Humprey, 1958*) needed by Montezuma quail to disperse.

Evidence suggests isolation by distance in Montezuma quail populations. Changes in temperature and precipitation patterns apparently expanded in altitude and latitude boundaries of deserts surrounding Montezuma quail habitat in the southwestern United States (*Archer & Predick, 2008*; *Seager et al., 2007*; *Williams et al., 2010*). Thus, constant

fragmentation, and predicted reduction of the extent of currently suitable habitats for the Montezuma quail (*Tanner et al., 2017*) may reduce corridors between populations, causing complete separation of populations and increase the risk of survival of the species in the mid and long term.

Given the little genetic structure found in this study and the resolution offered by DNA microsatellites, we were unable detect effects of Montezuma quail reintroductions from Arizona to Texas on genetic structure. There is no evidence that the reintroductions were successful and that any genetic structure was disrupted. However, we found a significant genetic difference between AZ and WTX. In addition, *Mathur et al. (2023)* also found that extant populations in Arizona and Texas are genetically distinct from one another, having diverged approximately 17,000 years ago.

The presence of isolated populations in mountain patches of habitat or "sky islands" is frequent along all edges of its geographic distribution, which may be embedded in arid, subtropical, or tropical vegetation. The existence of these isolated populations poses a conservation challenge for managers. These isolated populations have most likely differentiated, as those of numerous vertebrates inhabiting sky islands systems (*Barrowclough et al., 2006*; *Browne & Ferree, 2007*; *Hartley et al., 2023*; *Love et al., 2023*). The degree to which isolated Montezuma quail populations have differentiated deserves investigation. Montezuma quail in sky islands may have a similar degree of differentiation that led to the special management status of the Mount Graham red squirrel (*Tamiascirus fremonti grahamensis*) in Arizona (*U. S. Fish and Wildlife Service, 1987*). Furthermore, isolation in sky islands can also lead to local adaptation, which may also promote persistence of a species. Therefore, a genomic survey through the Montezuma quail geographic distribution is a research priority. Genome sequencing has been undertaken for the species (*Mathur et al., 2019*; *Mathur & DeWoody, 2021*; *Mathur et al., 2023*), but with minimal representation of Mexican and sky island populations. Genomic surveys will unveil genetic structure patterns relevant for conservation. For instance, translocations from one island population to another by federal and state game management agencies in the United States may have irreparably disrupted local genetics in such isolated populations. Like the Mount Graham's red squirrel, genomic studies of Montezuma quail populations in Texas, southeastern Arizona, and southwestern New Mexico sky islands might reveal that they may warranted special management status.

## CONCLUSIONS

The subtle genetic structure and low levels of genetic variation detected in our study is valuable for the future management of this charismatic game bird. This subtle genetic structure consists in clusters in Arizona, New Mexico, and Texas. Despite the differentiation, the analysis suggests some mixing between populations, which may indicate infrequent, long-distance dispersal, especially between New Mexico and Arizona populations. Migration between New Mexico and West Texas is also possible. In this context, our results support isolation by distance between populations, due to the non-independence between geographical distance and genetic distance. Montezuma quail populations in the southwestern United States may not be fully isolated, despite habitat

loss and fragmentation in this region as some corridors may open or close periodically depending on annual precipitation and its effect on herbaceous vegetation cover.

## ACKNOWLEDGEMENTS

We thank Robert Perez of the Texas Parks and Wildlife Department for his logistical support that made the collection of quail specimens in Texas possible. We also thank the volunteer hunters who collected or assisted in collecting specimens including Randy Gray, Steve Hopkins, Dennis Kavanagh, Zack May, Bill Miller, Mike Sullins, Peter Toot, Ray Trejo, Carlos Zuniga, and numerous anonymous quail hunters.

### Funding

This work was supported by the Arizona Quail Alliance, New Mexico Quail, Southern Arizona Quail Forever, Texas Parks and Wildlife Department, The Timken Foundation, and Pheasants Forever and Quail Forever in Texas. The funders had no role in study design, data collection and analysis, decision to publish, or preparation of the manuscript.

### Grant Disclosures

The following grant information was disclosed by the authors:
Arizona Quail Alliance.
New Mexico Quail.
Southern Arizona Quail Forever.
Texas Parks and Wildlife Department.
The Timken Foundation.
Pheasants Forever and Quail Forever in Texas.

### Competing Interests

The authors declare that they have no competing interests.

### Author Contributions

- Eduardo Sánchez-Murrieta conceived and designed the experiments, performed the experiments, analyzed the data, prepared figures and/or tables, authored or reviewed drafts of the article, and approved the final draft.
- Alberto Macías-Duarte conceived and designed the experiments, analyzed the data, prepared figures and/or tables, authored or reviewed drafts of the article, and approved the final draft.
- Reyna A. Castillo-Gámez conceived and designed the experiments, authored or reviewed drafts of the article, and approved the final draft.
- Alejandro Varela-Romero conceived and designed the experiments, performed the experiments, analyzed the data, authored or reviewed drafts of the article, and approved the final draft.
- Angel B. Montoya conceived and designed the experiments, authored or reviewed drafts of the article, and approved the final draft.

- James H. Weaver conceived and designed the experiments, authored or reviewed drafts of the article, and approved the final draft.
- Nohelia G. Pacheco-Hoyos performed the experiments, authored or reviewed drafts of the article, and approved the final draft.

### Field Study Permissions

The following information was supplied relating to field study approvals (*i.e.*, approving body and any reference numbers):

Tissue was extracted from specimens hunted in Arizona and New Mexico by numerous hunters under hunting licenses issued by Arizona Game and Fish Department and New Mexico Department of Game and Fish. Tissue samples from Texas originated from specimens collected under Scientific Permit Number SPR-0410-139 issued by Texas Parks and Wildlife Department.

### Data Availability

The individual genotypes (9 DNA microsatellite loci) for 119 Montezuma quail is available in the Supplemental File.

### Supplemental Information

Supplemental information for this article can be found online at http://dx.doi.org/10.7717/peerj.16585#supplemental-information.

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
