# Peer review of "Genetic variability and population structure of the Montezuma quail (Cyrtonyx montezumae) in the northern limit of its distribution"

_PeerJ, doi:10.7717/peerj.16585_

## Round 0.1 · original submission · Major Revisions

The reviewers disagreed with their recommendations, so I went with major revisions. I did a quick annotation of your manuscript for some general writing suggestions - by no means comprehensive. I would ask that you detail how you resolved the issues brought up by the reviewers, and in my annotated manuscript. While the writing was relatively clear, there is room for improvement. I comment on a variety of details in style, and you should see if you can find similar ways to improve the writing throughout the manuscript.

·

Basic reporting

The article is clear, unambiguous and technically correct. Literature cited is adequate and appropriate. The article is structured professionally and raw data are shared.

The article is descriptive in nature and does not need to present hypothesis.

Experimental design

Research question is clear, relevant and meaningful. Genetic information on this bird is previously lacking and this is the first to fill in and compare the genetic structure of this species. Methods are described clearly and use standard terminology and should be easily duplicated by others. Methods could have included larger samples from vertebrate museum specimens at many University and national collections. Perhaps future work would benefit from using more advanced, demanding genomic sequencing, but the methods used can still be very illuminating.

Validity of the findings

Discussion and conclusions are well integrated into existing literature. Underlying data are robust and treatment is sound. Conclusions are well stated given the methods and materials used to describe the genetic structure of the samples included.

Additional comments

This is an important start in understanding the genetic structure of a bird species that has evolved in recent times at the edge of a larger range in isolated, mountain islands of habitat. As such, these isolated populations have most likely differentiated, as many vertebrates have done on isolated islands. The degree to which they had differentiated needs to be more fully explored. The Montezuma Quail on each mountain island may have the kind of differentiation that, for example, lead to the management of the isolated Mt. Graham population of Tamiascirus fremonti grahamensis. This isolated squirrel population, at the edge of its range, has USFWS “endangered” status. It is entirely possible that the genetics of the Montezuma Quail (MQ) could reveal a similar status for the populations of MQ on the isolated mountain islands of Texas, SE Arizona and SW New Mexico. Climate change will only further limit the distribution of MQ. The authors should point this out. The populations of MQ on the sky islands are extremely unlikely to move between the mountain ranges as the quail have very specific habitats and specialize in diet of relatively few plants only found on the oak-savanna slopes of the mountain ranges.
This genetic study is suggestive. It is a good start. It relies on DNA microsatellites. The methods used in this study might be enhanced. Any study in today's world would  benefit from whole genome sequencing, which has been recently developed to expand on what could be learned from earlier DNA mircrosatellite studies.
The authors might pursue further study, or augmentation of this study, by contacting Professor Rauri Bowie at the Museum of Vertebrate Zoology (MVZ), University of California, Berkeley. Along with others, there is a major effort funded to sequence genomes for conservation planning. This larger project, California Conservation Genomics Project,(www.ccgproject.org) has produced sequences of the genome of California Quail and Mountain Quail- by Bowie’s lab. It is very likely the Bowie lab would welcome collaboration on the study of the genome of MQ. Having the other quail genomes already sequenced, means that MQ genomes could be processed for SNPs relatively quickly and with minimal costs.
The paper could be enhanced, or augmented with future papers, by including MQ genomes from museum specimens. There are over 430 specimens of MQ at various museums (CA, NM, TX, and larger US museum collections). MVZ has 131 specimens including specimens from Texas, Arizona, Chihuahua, Jalisco, Sinaloa, Michoacan, Coahuila, Aguascalientes, etc. MVZ has specimens from the Santa Rita, Huachuca, and SW NM. Gene sequences from the MQ in their main range in Mexico could be compared to a large sample of MQ from the isolated populations at their northern limits. Thus, the conservation or management needs of the mountain island MQ could be presented in perspective.
The paper is well done and should be published, but the authors should at least discuss the future need for genome sequencing to present the important larger pattern. The authors should also discuss how federal and state game management agencies that have moved individuals from one island population to another have often irreparably damaged local genetics in such isolated populations. Like the Mt. Graham’s Fremont squirrel, genomic studies of the mountain island MQ might reveal that they should not be hunted, but protected as unique and relatively rare birds.

Reviewer 2 ·

Basic reporting

1. The article is written in clear and understandable English.
2. The literature review is appropriate and complete, however, there is a previous work of great relevance, which should be cited and discussed. On the basis that a previous study, based on mitochondrial DNA, found that the phylogeographic structure also appears to be weak and populations from Arizona and Texas were not genetically distinct from one another.
Allen, T. H. (2003). Mitochondrial genetic variation among populations of Montezuma quail (Cyrtonyx montezumae) in the southwestern United States. Sul Ross State University.

3. Figures 1 and 3 should include the total distribution of the species, in order to have a global perspective that helps to understand the demographic and biogeographic history of the species. A search of GBIF records yields a structured pattern that could have a similarity to the findings here, the inclusion of those records could support your findings, while the inclusion of apparent relevant lowland barriers with rivers could be explored.

Figure 3 might contain the labels of the groups found by the STRUCTURE analysis: Arizona, WNM, CNM, and WTX.

Experimental design

1. In the first part of the results it is indicated that ten microsatellites were analyzed, however, it mentions that one was excluded, as indicated in Table 1 and in the summary. To avoid confusion, it should be worded in such a way that 10 were tested but 9 were used for the analysis.
2. Since the number of loci analyzed is very low, this should be considered in the discussion, pointing out the possible limitation of finding structure with so few loci.
3. It is not mentioned whether loci were evaluated for linkage disequilibrium.

Validity of the findings

One of the justifications for the analysis of the genetic structure of this species is that this information is essential for its management. It is also mentioned that as part of the management of the species, individuals have been translocated from Arizona to Texas. The article does not discuss the results obtained with the past management of this species, and does not give relevant suggestions for the management of this species.
The authors should address whether past translocation may have caused gene flow and whether this was reflected in the results obtained. In addition to giving specific suggestions for the management of the species, considering the results obtained.
The discussion could be enriched by considering the results obtained in the Spotted Owl, given that it is the same fragmented habitat, in addition to the fact that isolation by distance was observed in the genetic study. This reference could also be included in the introduction where it is suggested that habitat patches in a desert matrix could prevent gene flow. Barrowclough, G. F., Groth, J. G., Mertz, L. A., & Gutiérrez, R. J. (2006). Genetic structure of Mexican spotted owl (Strix occidentalis lucida) populations in a fragmented landscape. The Auk, 123(4), 1090-1102.


The discussion from line 224 through 230 should include the work of Tanner et al. (2019), which mentions that forward projections based on modeled climate data indicated a net loss for montezuma quail habitat
Tanner, E. P., Papeş, M., Elmore, R. D., Fuhlendorf, S. D., & Davis, C. A. (2017). Incorporating abundance information and guiding variable selection for climate-based ensemble forecasting of species' distributional shifts. PLoS One, 12(9), e0184316.

Additional comments

1. The year of the Arif & Khan, 2008 on line 51, must be 2009.


On line 173, the same observation of low diversity in peripheral populations is referred to by Mathur, et al. (2019), with previous work on this species, so it should be cited.

In line 181, indicate that the query is through Ebird, and indicate when the query was performed. As suggested, the inclusion of these Ebird and GBif observations in Figure 1 or 3 could give a context of the population genetic structure.

Line 85, add Central New Mexico

Reviewer 3 ·

Basic reporting

The papers used microsatellite loci to investigate population structure in Montezuma Quail. Overall, this is a relatively straight-forward paper. The analyses are appropriate, although a few aspects require clarification. In parts of the discussion, the authors overstate their results, and this aspect needs to be modified in a resubmission. I hope the authors find the comments below useful.

Introduction. This section of the paper needs extensive revision. I would start with outlining the biology, distribution and present management of the species and then move to why maintaining genetic diversity of metapopulations is important. However, isolation can also lead to local adaptation through selection, and this can be of importance to the persistence of a species. A more balanced discussion of the pros and cons of gene flow should be provided.

The raw data have been made available.

Experimental design

Methods.
- For scoring the microsatellite alleles, what size standard was used?
- Were some individuals scored more than once after being run independently on the sequencer? This would enable error rates associated with allele scoring to be quantified.
- The authors need to determine whether any of the loci are in linkage disequilibrium and if so, those loci should be excluded.

Validity of the findings

Results.
Line 137 – instead of exclusive say “private alleles”

I would present figure 3 as bar-chart for each individual. The authors are misunderstanding how this method works. When you look at the pie-charts which show the genetic composition of each cluster, there looks to be equal contributions from each one (green, red, and blue). This is indicative of panmixia not genetic structure. The Evanno method is recovering a delta K of 3 because it cannot figure out how to partition the allelic diversity, and when you look at the bar-graphs you will see that there is no sensible geographical clustering. This result is also consistent with the weak differentiation as reflected by Rst.

Discussion
The first three sentences of the discussion need to be reworked. How do you know that an Ho of 0.22 +/- 0.02 is low for the species – the authors need to provide a meaningful comparison to place this in context.

To what populations does the Mathur et al. 2019 paper refer?

What is the context for the sentence about the Odontophoridae on lines 166-167? I do not follow how this connects with the sentences above.

Genetic structure. Much of this is pure speculation. Your results suggest very little genetic structure and that what there is, is due to isolation-by-distance. This is what I would focus on and what I would argue would be expected in a situation where birds have been translocated. With the exception of putative isolation-by-distance, there is just not enough power with the microsatellite data to say any other process is operating. What is really needed is whole-genome sequencing to tease apart demography, gene flow, adaption and drift, in order to be able to implement an effective conservation management plan.

Lines 212-215 – this is not correct. Your results suggest very weak genetic structure and the significant Rst values are likely not biologically relevant but rather an artifact of a frequency-based statistical approach.

Lines 221-223. Your results do not provide any evidence of a ring species, you simply have evidence of gene flow either currently taking place or recently having stopped.

---

## Round 0.2 · Minor Revisions

I appreciate your effort in improving the quality of your manuscript. I read through it myself again, and am providing you with an annotated copy of your manuscript with many suggestions to improve the quality of the writing and the presentation.

Please read carefully my comments in the annotated version. Here I will simply point out a few details. You abbreviate west Texas and the two New Mexico sites, however, because you only have one site in Texas, and your map and description explain where it is, you might simply call it the Texas population. Thus, you have the AZ population, the TX population and the two New Mexico populations (CN and WN, perhaps, and they will all have 2-letter abbreviations - not really too important. Also, you abbreviate Montezuma Quail as Montezuma, rather than quail. Quail has a benefit in being both singular and plural. In any case, you could abbreviate as quail. Whatever you decide, you might not use abbreviations in the Discussion as it will make the writing easier to read. I'm not that concerned about this issue, but just thought I'd mention it for you to consider.

Next, there are some issues with your correlation value and the associated P value. Double-check those please. And, you might improve your figures a wee bit. See my suggestions.

I hope you find these comments and suggestions helpful.

---

## Round 0.3 · Minor Revisions

I appreciate your efforts - this manuscript is quite improved. The comments I have in the annotated manuscript are minimal, really, but I thought important enough to consider. If you disagree with me, feel free to explain your point of view. My main observations have only to do with Figures 1 and 3 (and how they are described in the text, although I only comment in the figures).

---

## Round 0.4 · accepted · Accept

I think each version has improved, so thank you for the effort. I'd still like to comment on Figure 3. If you decide to follow my suggestion, I am sure that it will require minimal text adjustment and only adjustment to the figure.

In the new legend you state: "...showing suggestive evidence of a longitudinal gradient in genetic differentiation." yet, nothing in the figure is associated with longitude, so how does it suggest that? Also, the sentence would be better worded as "....suggesting evidence...." ("showing suggestive" is poorly worded). I think something about longitude needs to be made clear in the legend and perhaps the text.

Next, the point of this figure seems to be to show how well regions are indicated by genotype/ancestry. That being the case, it should clearly show how well Arizona is represented IN Arizona (in other words, by descending purple), and how well New Mexico is represented IN New Mexico (and so, by descending blue), and of course, Texas in Texas (descending yellow). In your rebuttal letter, the alternative was much worse, but not what I was suggesting - in fact, I'm sure I'm not understanding what ordering along the X-axis you used. In any case, the way I am suggesting allows the reader to figure out how well location indicates ancestry. At any rate, you can decide.